# Vector Quantization for Reversed Disease Progression: Further Investigations

**Chih-Chieh Chen**[1]                                             JACKFRANK@GMAIL.COM
**Chang-Fu Kuo**[1,2,3]                                            ZANDIS@GMAIL.COM

[1] *Center for Artificial Intelligence in Medicine, Chang Gung Memorial Hospital, Taoyuan, Taiwan*

[2] *Medical Education Department, Chang Gung Memorial Hospital, Taoyuan, Taiwan*

[3] *Division of Rheumatology, Allergy and Immunology, Chang Gung Memorial Hospital, Taoyuan, Taiwan*

**Editors:** Under Review for MIDL 2026

## Abstract

Interpretability plays a pivotal role in the collaboration between artificial intelligence (AI) systems and clinicians. It enables clinicians to critically reassess the rationale underlying AI-generated predictions. Moreover, translating these interpretations into clinically meaningful quantifications is feasible even for more granular algorithms, thereby potentially reducing the extensive annotation efforts typically required. Recently, a novel approach was introduced to generate reversed disease progression trajectories by applying conditional flow matching within the latent space of an autoencoder, jointly training a linear classifier. However, the architectural design, training procedures, and objective functions associated with the flow matching network warrant further investigation and refinement. In the present study, we implement this concept utilizing a recently proposed vector-quantized autoencoder framework incorporating Sinkhorn-based quantization. Our findings indicate that reversed disease progression can be consistently generated even in the absence of joint classifier training. Additionally, the method preserves strong spatial correspondences between the pixel domain and latent representations, enabling the synthesis of desired images through a CutMix-inspired algorithm. We demonstrate the efficacy of our approach by applying it to the weakly supervised quantization of midline shift distances.

**Keywords:** Interpretable AI, Medical Image, Clinical Diagnosis.

## 1. Introduction

The clinical context presents several distinctive challenges. Firstly, there exists an extensive volume of data, the majority of which is unlabeled and can only be derived from clinical reports. The generation of fine-grained annotations necessitates specialized domain knowledge, rendering such annotations largely unavailable in most cases. This issue is further exacerbated in the case of rare diseases, where each institution typically possesses only a few hundred annotated instances. Under these circumstances, training a robust classifier or generative model becomes particularly difficult (Karras et al., 2020; Kadkhodaie et al.). Secondly, the requirement for interpretability in clinical settings is generally more stringent than in other fields. Clinicians prefer model interpretations that are comprehensible to facilitate the reexamination of predictions. Moreover, it is advantageous to provide quantifiable information that can be integrated into various scoring systems, thereby alleviating the workload of clinical practitioners.

In this study, we aim to concentrate on measuring the midline shift distance (Broder, 2010) , with the goal of developing an interpretable approach. Following the concept proposed by (Chen et al., 2019) to construct a transparent reasoning process, recently (Chih-Chieh and Chang-Fu, 2025) introduced a method for quantifying disease progression by generating a reversed trajectory, specifically transforming pathological images back to their normal counterparts, as depicted in Fig. 1. Their approach employs conditional flow matching within the latent space of an autoencoder, which is concurrently trained alongside a linear classifier. Although this concept demonstrates considerable potential, certain aspects require correction and warrant more comprehensive examination.

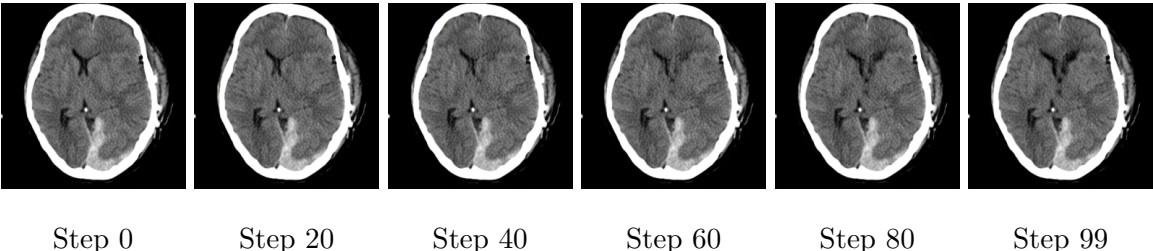

Step 0    Step 20    Step 40    Step 60    Step 80    Step 99

Figure 1: Reversed disease progression generated by our proposed architecture.

The authors of (Chih-Chieh and Chang-Fu, 2025) introduced an auxiliary classification head within the latent space of the autoencoder, which is trained jointly to distinguish between class-related and class-unrelated image tokens. For broader applicability, it would be advantageous to train the autoencoder and the classifier independently; specifically, one could train the autoencoder on a large dataset and subsequently train a classifier using images associated with specific target pathologies.

Moreover, the flow matching model faces challenges in regulating irrelevant features when such features are more prevalent in images from the source domain compared to those from the target domain. This issue is particularly common in clinical contexts. In our case, midline shift arises as a stress response to brain hematoma; however, not all patients with brain hematoma exhibit midline shifts. Moreover, it is generally easier to obtain images lacking both midline shift and brain hematoma than to acquire images without midline shift but with brain hematoma. Consequently, during the transfer of images to the target domain, these features may inadvertently be transferred alongside other characteristics.

In this study, we aim to examine these challenges and explore strategies to enhance performance. To generate more realistic images, in this work we work on a more fine-grained VQ model. Our investigation builds upon the recently developed OptVQ framework (Zhang et al., 2024), which attains a high codebook utilization rate by relaxing the vector quantization (VQ) objective—specifically, by minimizing the distances between latent vectors and their nearest image tokens—through the application of a Sinkhorn-like algorithm. Notably, we observed that when employing OptVQ, pathological images can often be precisely transformed into normal images, even in cases exhibiting minimal midline shift distances. To address the issue of data imbalance, we discovered a pronounced spatial correspondence between pixel space and the latent space within vector-quantized autoencoders. In particular, we found that applying a CutMix-inspired algorithm (Yun et al., 2019) within the latent space frequently yields realistic reconstructed images. Motivated by this observation, we

propose a novel approach that involves substituting segments of the latent codes of target images with those from source images at locations characterized by high pixel intensity. We refer to this method as intensity shuffling.

We also aim to examine the underlying factors contributing to the enhanced performance observed when employing Sinkhorn-based quantization. Our analysis reveals that OptVQ exhibits a more pronounced spatial correlation between the pixel domain and the latent representation, as it effectively preserves local patches even when there is significant noise in adjacent regions. Motivated by this finding, we propose a method to transform images exhibiting midline shifts into normal images by substituting the cropped regions corresponding to the frontal horns with those from normal images. We refer to this approach as patch shuffling.

In summary, the contributions of this study are as follows:

- We elucidate the technical challenges associated with simulating the reverse progression of disease as introduced by (Chih-Chieh and Chang-Fu, 2025).

- To enhance the quality of generated image sequences, we apply the same conceptual framework to the recently proposed OptVQ model, achieving notable improvements.

- We conduct a detailed investigation into the role of Sinkhorn-based quantization methods and introduce two cost-effective data augmentation techniques—intensity shuffling and patch shuffling—designed to address data imbalance issues and facilitate the transformation of pathological images into normal ones, respectively.

- We evaluate the performance of our proposed methods using the midline shift dataset to demonstrate their efficacy.

## 2. Related Works

### 2.1. Basics for Midline Shift

In accordance with the methodology presented by Liao et al. (Liao et al., 2018), this subsection provides a concise overview of the concept of midline shift distance. The ideal midline, depicted as the blue line in Fig. 2 (a), is defined as the straight line connecting the anterior and posterior falx cerebri. Conversely, the actual midline, represented by the red curve, corresponds to the anatomical boundary that genuinely separates the left and right cerebral hemispheres. In a neurologically normal individual, these two lines are expected to be closely aligned. However, pathological conditions such as stroke, hematoma, or other brain injuries can induce increased intracranial pressure, resulting in deformation of the actual midline. The quantification of this intracranial pressure can be achieved by measuring the midline shift distance (MLS), which is defined as the displacement between the ideal and actual midlines at the level of the foramen of Monro (FM)—the anatomical channel connecting the frontal horns of the third ventricle.

### 2.2. Sinkhorn based Vector Quantization

(Zhang et al., 2024) proposed OptVQ, a method inspired by the optimal transport theory. Instead of replacing each embedding vector by the closest codebook vector, (Zhang et al.,

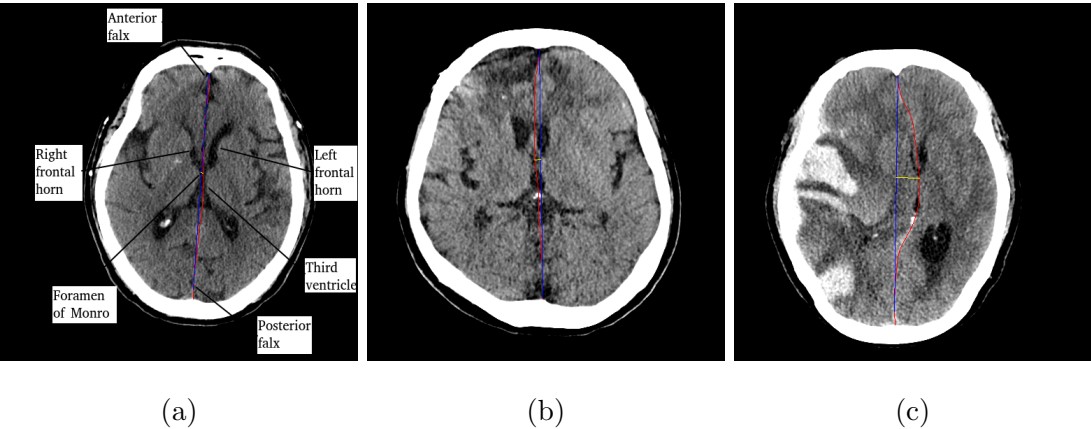

Figure 2: Illustrations of CT images (a) without midline shift, (b) with midline shift distance $<= 5$ mm (c) with midline shift distance $> 5$ mm.

2024) proposed to view each batch of latent embeddings $z_e(x)$ and the codebook $e$ as two distributions of vectors, and transport $z_e(x)$ to $e$ with minimal 2-Wasserstein distance, i.e., the distance in Equation 7. Since the marginal of $\pi$ on $e$ is exactly the probabilistic distribution of $e$, which means that every code vector of $e$ will possibly be assigned by one of the embedding vector in $z_e(x)$, hence the utility rate of $e$ is increased. Computationally, Sinhorn algorithm (Peyré et al., 2019) is adopted for approximating the transport map: given a small number $\epsilon > 0$, initializing the assignment transport matrix $C$ as the pairwise similarity matrix between pairs of vectors in $z_e(x)$ and $e$. Let $u = 1 \in \mathbf{R}^K, v = 1 \in \mathbf{R}^L$, where $L, D$ are the sizes of $z_e(x), e$, respectively. Let $K = e^{-\frac{C}{\epsilon}}$, at each iteration, $u, v$ are updated as the following way:

$$u \leftarrow \frac{z_e(x)}{Kv}, v \leftarrow \frac{e}{Ku}. \tag{1}$$

The assignment matrix will be given by $\mathrm{diag}(u)K\mathrm{diag}(v)$, where $\mathrm{diag}(u), \mathrm{diag}(v)$ are the diagonal matrices in $\mathbf{R}^{K \times K}, \mathbf{R}^{L \times L}$, respectively. From Remark 4.6 of (Peyré et al., 2019), $O(\frac{n^2 \log(n)}{\epsilon^3})$ iterations are enough to achieve $\epsilon$-approximation, where $n = \max(K, L)$. In the work (Zhang et al., 2024), $\epsilon = 1$ and the authors only use 3 Sinkhorn iterations. However, empirically the authors demonstrated the codebook utility rates are already close to 100% in all the experiments. In our work, we adopt OptVQ as our autoencoder.

## 3. Method

### 3.1. Investigate the Trade-off between Visual and Quantitative Qualities

Our objective is to develop an interpretable model. Specifically, when given a CT image that may show disease, our model first converts it into a normal image and then extracts quantitative information using various metrics. We aim for the generated normal image to not only enable highly accurate quantification but also remain unbiased. As demonstrated in Section 2.1, images exhibiting midline shift distance are typically associated with stroke

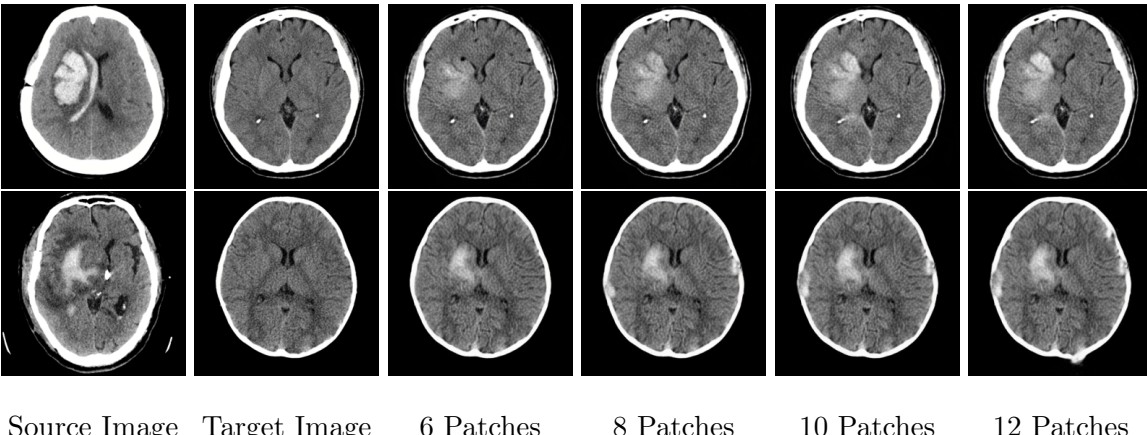

|                |                |           |           |            |            |
|:--------------:|:--------------:|:---------:|:---------:|:----------:|:----------:|
| Source Image   | Target Image   | 6 Patches | 8 Patches | 10 Patches | 12 Patches |

Figure 3: Illustrations of the effects of intensity shuffling on OptVQ with different numbers of patches.

or hematoma. Nevertheless, in certain instances, the midline shift is not attributable to these conditions or is only partially caused by them. Conversely, empirical observations indicate that the majority of the collected normal CT scans do not present with stroke or hematoma. Given that our approach employs conditional flow matching to transform pathological images into normal images, it follows that most features related to hematoma are eliminated, including regions that do not contribute to the midline shift. For instance, in the source image of the first row depicted in Fig. 3, there are two hemorrhagic components; however, only the component adjacent to the frontal horns can be considered the causal factor for the midline shift. Despite this, we observed that the flow matching model tends to remove all hemorrhagic regions, likely because the training dataset contains very few normal images exhibiting hemorrhage.

To relieve the issue, in the following subsections we propose different kinds of solutions and test the visual and quantitative performances in the next section.

### 3.2. Proposal 1: Reducing Bias through Regularization at the Training Time

To address this challenge, we observed that although the default configurations of VQ-GAN and OptVQ incorporate attention layers within the bottleneck layers of both the encoder and decoder, a pronounced spatial correspondence persists between the input image patches and the latent vectors at their respective locations. Consequently, as depicted in Fig. 3, we implemented a data augmentation technique whereby segments of patches corresponding to hemorrhage locations are directly shuffled during the training of the flow matching model. We refer to this augmentation approach as intensity shuffling.

### 3.3. Proposal 2: Reducing Bias through Adding Deformation Bias

Inspired from (Joshi and Hong, 2023), (Chih-Chieh and Chang-Fu, 2025) modeled the reverse disease progression as a metamorphic image registration task. To achieve this, they introduced a shallow deformation layer on top of the standard UNet outputs, which trans-

forms the intensity outputs to produce the final warped results. Both the spatial and deformation layers, which consist of three ResBlocks, are kept shallow to improve their interaction. This constraint encourages the neural network to learn where to apply the deformation layer. This adapted UNet design is referred to as UNetWarp.

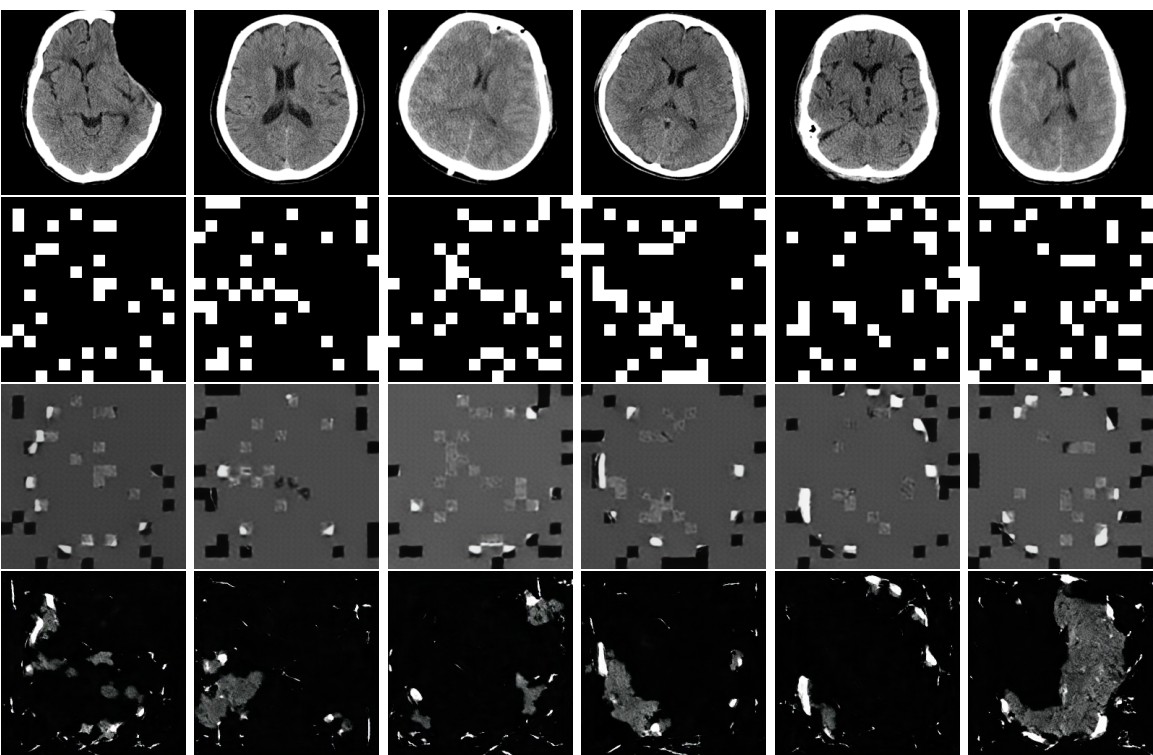

Figure 4: Illustrations of the reconstructions of masked images. From above to below: original images, masks, reconstructed images by OptVQ, reconstructed images by VQGAN.

## 3.4. Masked Image Reconstruction as a Test of Spatial Correlation

Conceptually, Sinkhorn-based quantization differs from nearest neighbor-based quantization in that the final quantized value of each patch is influenced not only by the patch itself but also by its correlations with other patches within the same image. To empirically validate this hypothesis, and drawing inspiration from the works of (He et al., 2022; Li et al., 2022), we replaced 85% of the patches with the average pixel value of the original image and evaluated the reconstruction performance. As depicted in Fig. 4, the VQGAN model frequently produces fragmented patches under these conditions. In contrast, the OptVQ model reconstructs the patches at the unmasked locations with notable fidelity. Although similar reconstruction quality can be achieved using Variational Autoencoders (VAE) or other vector quantization techniques (refer to Fig. 12 and Fig. 15 for further details), we observed an intriguing phenomenon: for patches in masked regions, VQGAN and VAE tend to generate patches with darker coloration, potentially treating these patches as outliers.

Conversely, OptVQ generates patches whose color closely approximates the average value of the original image. While the implications of this observation warrant further investigation, it suggests that OptVQ may possess superior capabilities in reconstructing unseen or outlier data. This insight motivated the introduction of patch shuffling, a CutMix-like data augmentation method, which is discussed in the subsequent subsection.

### 3.5. Proposal 3: Postprocessing by Patch Shuffling

As demonstrated in Fig. 4, our findings indicate that OptVQ is capable of reconstructing patches even when the neighboring patches are unrelated or untrained. In our study, all images were rigidly registered, ensuring that the locations of the frontal horns remain proximate, even in images exhibiting midline shifts. This experiment, depicted in Fig. 4 prompted us to investigate the following question: if patches are cropped from the frontal horn regions of one latent code and subsequently pasted onto another latent code, will the resulting generated image be semantically and perceptually coherent? The results for OptVQ, presented in Fig. 5, reveal that the method reconstructs the frontal horns with near-perfect accuracy and produces minimal artifacts at the boundaries of the cropped regions. For comparative purposes, the corresponding results obtained using a VAE are shown in Fig. 11.

This form of data augmentation is referred to as patch shuffling. Notably, it is possible to directly produce plausible images without employing a flow matching model (refer to Appendix C for further details). In the present study, during the inference phase, patch shuffling is applied to the input and generated images to ensure that unrelated features remain constant throughout the reversed disease progression generation process.

### 4. Experiments

The overall procedure is summarized in Fig. 6. The training and testing datasets were from the Chang Gung Research Database (CGRD) (Tsai et al., 2017). For the training data, we used 1177 slices with midline shifts and 2221 normal slices from 907 Brain CT images. During the training of the flow matching network, the slices are further rigid registered to minimize rotational discrepancies involved in the reversed disease progression path. For testing data, we used 346 normal slices (refered to "normal" in Table 1), 178 slices with midline shift distances ranging from $2mm$ and $5mm$ (refered to "small MLS" in Table 1), and 119 slices with midline shift distances greater than $5mm$ (refered to "large MLS" in Table 1). To further quantify our results, we apply diffeomorphic registration to measure the displacement between the original and transformed images, which serves as an estimate of the midline shift distance of the original image. All the experiments were conducted with single Nvidia V100 GPU. For the flow matching model, we adopt OT-CFM (Tong et al., 2023) as the training objective and train the models for 20000 steps with a batch size of 80, using the Adam optimizer with a learning rate $2e - 4$. For intensity shuffling, during each iteration, we choose patches with the top-k highest intensity values, where k is randomly selected from a uniform distribution between 10 and 20. We employ the B-spline diffeomorphic registration technique implemented in AirLab (Sandkühler et al., 2018) to register the input images with the generated images. The resulting deformation fields are utilized to estimate the midline shift distance.

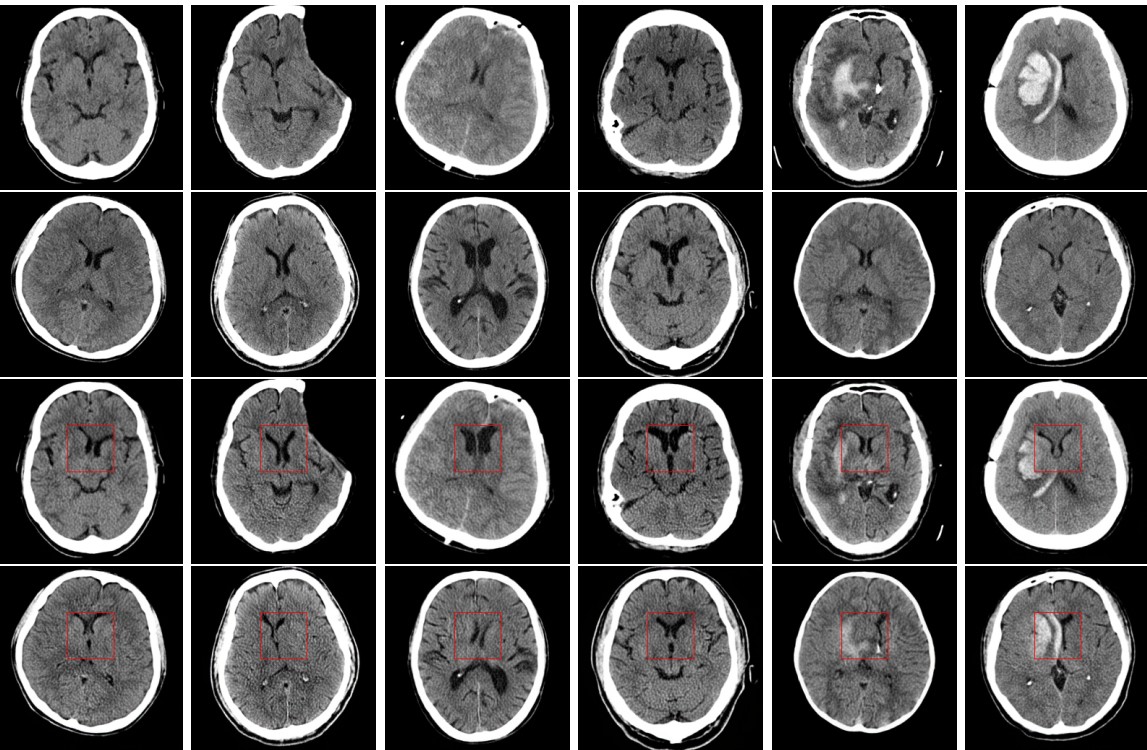

Figure 5: Illustrations of the patch shuffling. From above to below: source images, target images, source images after patch shuffling with target images, target images after patch shuffling with source images.

Step 1: Training an autoencoder for the flow matching model in the latent space.

Step 2: Training a vector field using OT-CFM and either augment on the training pair or add warped bias in the network architecture.

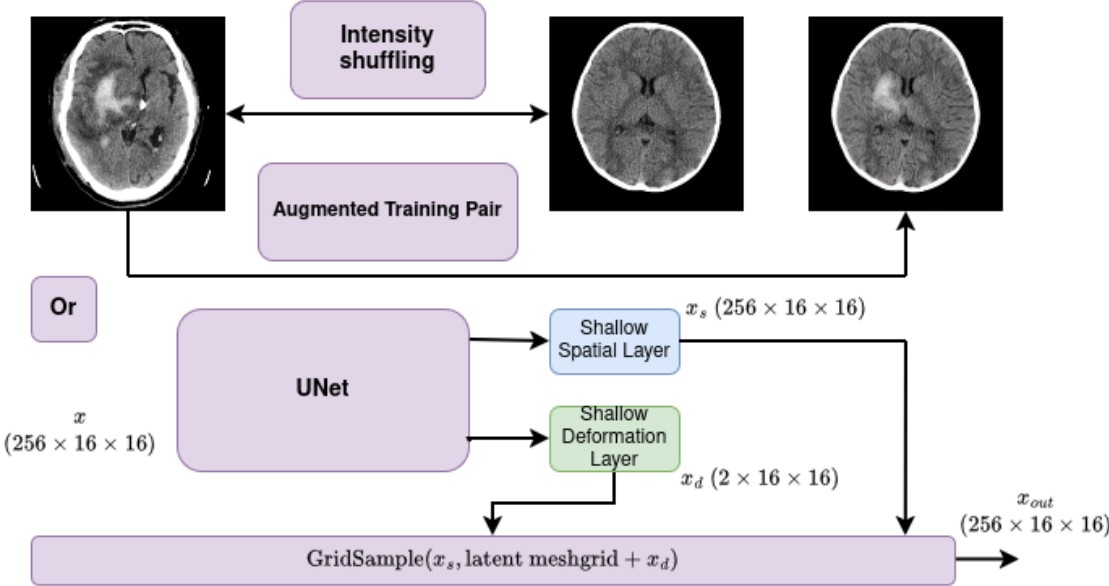

Intensity shuffling

Augmented Training Pair

Or

$x$
$(256 \times 16 \times 16)$

UNet

Shallow Spatial Layer — $x_s$ $(256 \times 16 \times 16)$

Shallow Deformation Layer — $x_d$ $(2 \times 16 \times 16)$

$x_{out}$
$(256 \times 16 \times 16)$

$\mathrm{GridSample}(x_s, \text{latent meshgrid} + x_d)$

Step 3: Inference possibly with patch shuffling to further fix the features outside the target regions

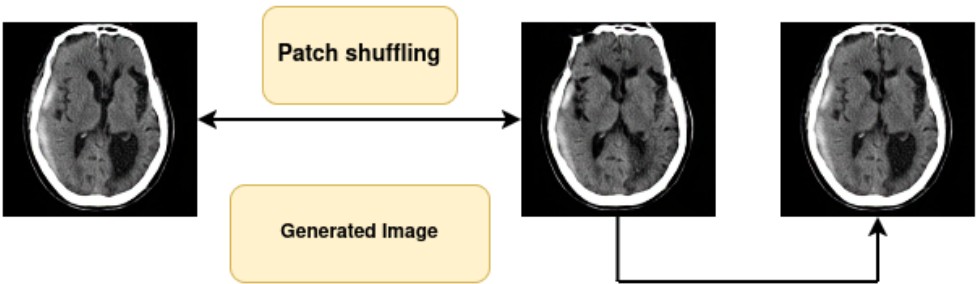

Patch shuffling

Generated Image

Step 4: Quantification by image registraion

Figure 6: Illustrations of the overall procedure.

To test the trade-off between visual quality and quantitative performance across different combinations of architectures and methods, we established several baseline approaches. The simplest baseline involves registering all potentially diseased CT images to a fixed template. We also assessed performance using a variational autoencoder as described in (Rombach et al., 2022). For the UNet model applied to the latent space of OptVQ, we tested OT-CFM in three variations: without any proposed augmentations, with intensity shuffling, and with both intensity and patch shuffling. For UNetWarp, we evaluated OT-CFM without augmentations and with patch shuffling.

The quantitative results are summarized in Table 1 and the visualization results are illustrated in Fig. 7. We would like to compare these results to the results reported in (Chih-Chieh and Chang-Fu, 2025), report mean absolute errors (MAEs) of 1.19 mm for normal MLS, 2.59 mm for small MLS, and 4.23 mm for large MLS. For registration with a fixed template, it is not surprising that the mean absolute errors (MAE) on normal images is large, because the positions of the frontal horns vary among normal images, we refer to the first three CT images in the first column of Fig. 7 for details. For variational autoencoder, the MAE on the normal case is also high. As shown in the second column of Fig. 7, we see that the tiny and distinct features have vanished in the generated outputs. Additionally, the generated frontal horns tend to be thicker compared to those in the original images. it is surprising that the MAE for both normal images and those with significant MLS remains relatively low. From the first 3 CT images in the third column of Fig. 7 we see that the differences between the source image and the generated image near the frontal horns are are negligible. However, as shown in the last three CT images in the same column, important features such as edema and hematoma are often overlooked. On the contrary, when applying the intensity shuffling, there are always some slight changes around the frontal horns, which could explain why the MAE for normal cases is higher than that of OT-CFM without any augmentations. However, at a certain level, pathological features are not completely lost. When combined with patch shuffling, nearly all pathological features located away from the frontal horns are retained. UNetWarp works surprisingly well for normal imagesachieving the lowest mean absolute error (MAE) among all methods. Additionally, it retains more pathological details compared to the UNet model. However, for images with significant midline shift, as seen in the last two examples in the fifth row of Fig. 7, some artifacts may appear. Combining UNetWarp with patch shuffling partially addresses the problem, but it also reduces accuracy in cases involving large MLS.

To illustrate how intensity shuffling and patch shuffling create more unbiased trajectories, we conducted a simple experiment to measure how well flow matching models preserve hemorrhagic regions. Specifically, we selected CT images containing large acute hematomas by counting the number of pixels with high intensity values. For these selected slices, we created binary masks representing these pixels. We then generated similar masks from the outputs of various flow matching models and calculated their mean Intersection over Union (mIoU) scores. The results are summarized in Table 2. Unsurprisingly, intensity shuffling + patch shuffling achieves the best performance.

Some of these errors may be attributed to unsuccessful image generation. However, we observe that numerous cases, particularly those involving hemorrhage, edema, or intensity variations, demonstrate inconsistencies between the deformation fields and the actual reversed progression of the disease. These instances are detailed in Appendix J.

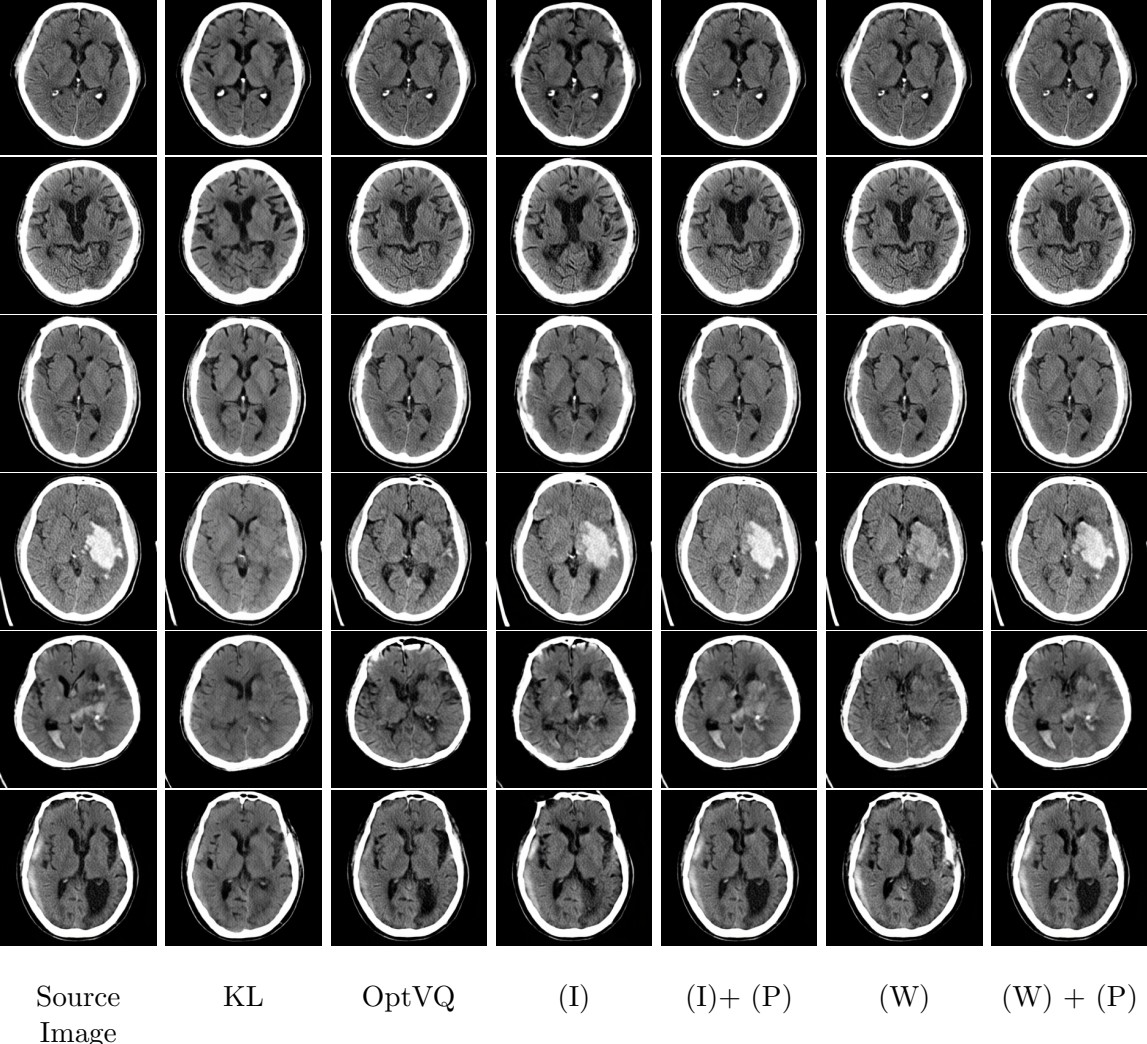

|  Source Image | KL | OptVQ | (I) | (I)+ (P) | (W) | (W) + (P) |

Figure 7: Illustrations of the generated results for different methods. (I) refers to intensity shuffling, (P) refers to patch shuffling, (KL) refers to use variational autoencoder, and (W) refers to use the UNetWarp network.

Table 1: Mean absolute errors of unsupervised midline shift quantizations. (I) refers to intensity shuffling, (P) refers to patch shuffling, and (KL) refers to use variational autoencoder.

|  | Normal | Small MLS | Large MLS | AVG (mm) |
|---|---|---|---|---|
| Fixed template | $8.99 \pm 0.41$ | $7.66 \pm 0.57$ | $4.89 \pm 0.62$ | $7.86 \pm 0.29$ |
| KL + UNet | $2.84 \pm 0.21$ | $2.96 \pm 0.40$ | $\mathbf{4.37 \pm 0.66}$ | $3.16 \pm 0.19$ |
| OptVQ + UNet | $0.85 \pm 0.11$ | $2.53 \pm 0.36$ | $4.48 \pm 0.63$ | $1.99 \pm 0.15$ |
| OptVQ + UNet + (I) | $1.54 \pm 0.21$ | $2.26 \pm 0.27$ | $5.12 \pm 0.70$ | $2.40 \pm 0.20$ |
| OptVQ + UNet + (I) + (P) | $1.41 \pm 0.19$ | $\mathbf{2.09 \pm 0.24}$ | $5.19 \pm 0.77$ | $2.30 \pm 0.17$ |
| OptVQ + UNetWarp | $0.66 \pm 0.08$ | $2.14 \pm 0.28$ | $4.58 \pm 0.66$ | $\mathbf{1.79 \pm 0.12}$ |
| OptVQ + UNetWarp + (P) | $\mathbf{0.64 \pm 0.07}$ | $2.24 \pm 0.25$ | $5.26 \pm 0.71$ | $1.94 \pm 0.12$ |

Table 2: Estimations of mIoU of acute hemorrhagic regions in each case.

| KL | OptVQ | (I) | (I) + (P) | (W) | (W) +(P) |
|---|---|---|---|---|---|
| $0.62 \pm 0.02$ | $0.70 \pm 0.02$ | $0.86 \pm 0.07$ | $\mathbf{0.99 \pm 0.02}$ | $0.81 \pm 0.02$ | $0.97 \pm 0.05$ |

## 5. Conclusion

In the present study, we conduct a comprehensive analysis to determine the extent to which generalization arises from the autoencoder component itself. Specifically, we focus on the recently introduced OptVQ model, which employs Sinkhorn-based quantization to produce image tokens. Our findings indicate that OptVQ is capable of reconstructing images with minimal artifacts when latent vectors are augmented using a CutMix-inspired augmentation technique. Motivated by this observation, we propose two novel augmentation strategies: intensity shuffling and patch shuffling. These approaches can be regarded purely as methods for generating novel images and have the potential to enhance the performance of conditional flow matching generation.

Given the limited number of images exhibiting hemorrhage in the collected dataset, intensity shuffling is utilized to augment normal images during the training of the flow matching model. Additionally, when reconstructing the reverse disease progression, it is preferable to maintain unrelated features unchanged; however, controlling the flow matching model to achieve this proves challenging, as minor alterations occur even in features that are entirely unrelated. To address this challenge, we augment the input images with generated images produced through patch shuffling applied to the regions corresponding to the frontal horns. Our findings indicate that the generated images exhibit minimal artifacts, and all unrelated features remain stable, with some compromise on accuracy.

We demonstrated several potential combinations of methods to explore the balance between visual and quantitative quality, aiming for more realistic and unbiased trajectory generation. The mean absolute errors in our study remain high and are not yet comparable to fully supervised methods. For future research, we plan to explore ways to further reduce these gaps.

## Acknowledgments

Both authors acknowledge funding from the Center for Artificial Intelligence in Medicine at Chang Gung Memorial Hospital, via grant agreement no. CLRPG3H0016 and no. CORPG3L0463

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

## Appendix A. Related Works for Quantifying Midline Shift Distances

(Liao et al., 2010) developed a landmark-based automatic midline shift measurement. Deep learning based methods are proposed by (Pisov et al., 2019; Wang et al., 2020; Wei et al., 2019; Nguyen et al., 2021) to handle the extreme cases which rule-based methods do not work well. (Pisov et al., 2019; Wang et al., 2020; Wei et al., 2019) transferred the midline shift measurement problem as the midline segmentation problem. (Nguyen et al., 2021) applied 2D-UNet and 3D-UNet to segment the anterior falx, posterior falx, and the foramen of of Monro. Where the performances are promising, lots of annotations are required.

(Gong et al., 2023) proposed a semi-supervised learning approach: Instead of only outputting a single midline shift distance value, the authors attempted to predict a deformation field and predict the midline shift distance as the maximal deformation distance. The annotations for each image are the location where clinicians measure the midline shift distance, and the midline shift distance. The authors combined labeled and unlabeled images to train a class-conditional diffusion model. For labeled images, the authors first used classifier-free guidance (Ho and Salimans, 2021) to generate the image guided by the normal class label. The original and generated images are combined as the input of their designed deformation network to generate predicted the deformation field.

Our approach is similar to (Gong et al., 2023). Instead of implicitly using the generated images as input features to train a deformation field, we directly compared the original image and generated image, and generate the deformation field by image registration.

## Appendix B. Pseudocode for the Intensity Shuffling

The pseudocode for the intensity shuffling is presented in Algorithm 1.

---
**Algorithm 1:** Pseudocode for Intensity Shuffling

---
**Input:** Image pair $x_0, x_1$. Encoder $Enc$. Pixel intensity threshold $t_{min}, t_{max}$. Number of pixels for shuffling $n$. Indicator function for the top-$k$ values of tensors in the spatial dimensions Select_Mask.

**Output:** Augmented latent vector $x_1^e$

$x_0^e \leftarrow Enc(x_0), \; x_1^e \leftarrow Enc(x_1)$

$h \leftarrow x_0^e.\text{size}()[-2], \; w \leftarrow x_0^e.\text{size}()[-1]$

$x_0 \leftarrow \text{torch.mean}(x_0, \dim = 1).\text{unsqueeze}(1)$

$x_0 \leftarrow \text{torch.nn.functional.adaptive\_avg\_pool2d}(x_0.\text{float}(), (h, w))$

$x_0 \leftarrow \text{torch.where}(x_0 > t_{min}, 1, 0) \cdot \text{torch.where}(x_0 < t_{max}, 1, 0)$

$x_0 \leftarrow \text{Select\_Mask}(x_0, n)$

$x_1^e \leftarrow x_0 \cdot x_0^e + (1 - x_0) \cdot x_1^e$

**return** $x_1^e$

---

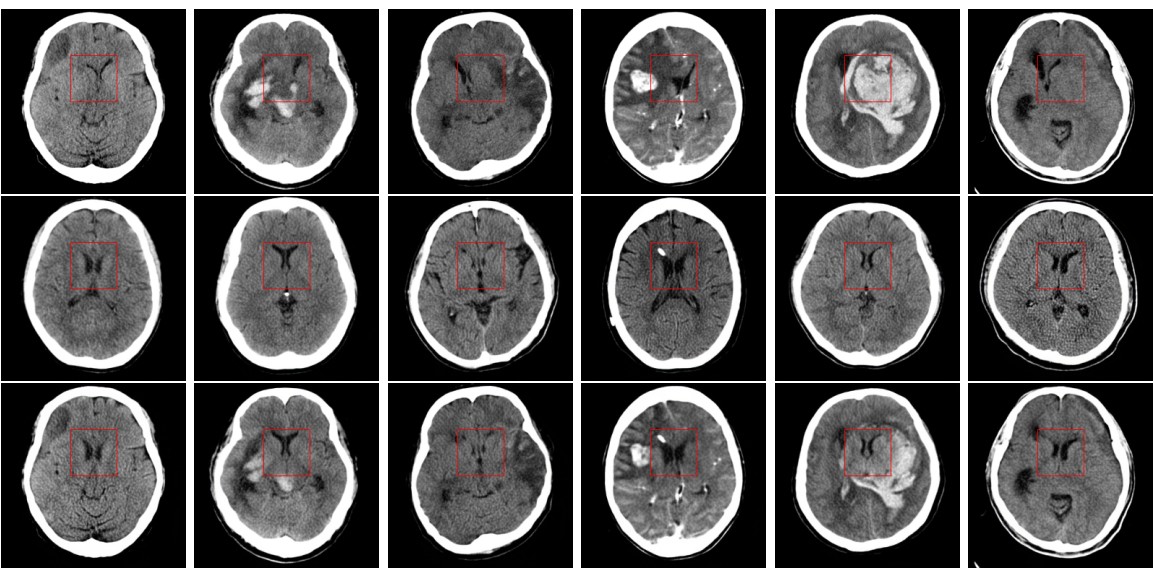

Figure 8: Illustrations of the generations of reverse disease progression by purely using the patch shuffling.

## Appendix C. Pseudocode for the Patch Shuffling

Algorithm 2 outlines the pseudocode for the patch shuffling procedure. Notably, the application of Algorithm 2 alone can occasionally produce images that are semantically coherent, as demonstrated in the first two columns of Fig. 8. However, for images exhibiting substantial midline shift distances, minor semantic inaccuracies tend to occur, as observed in the last four columns of Fig. 8.

---

**Algorithm 2:** Pseudocode for the Patch Shuffling

---

**Input:** latent code pair $x_0, x_1$. Crop coordinates $h_{min}, h_{max}, w_{min}, w_{max}$.
**Output:** Augmented latent vector $x_0^f$
$M \leftarrow torch.zeros\_like(x0)$ , $M[h_{min} : h_{max}, w_{min} : w_{max}] = 1$
$x_0^m \leftarrow M \cdot x_0$, $x_1^m \leftarrow M \cdot x_1$
$x_0^m \leftarrow torch.flatten(x_0^m, start\_dim = 1), x_1^m \leftarrow torch.flatten(x_1^m, start\_dim = 1)$
$d \leftarrow torch.cdist(x_0^m, x_1^m)$
$d \leftarrow torch.argmin(d, dim = 1)$
$x_1 \leftarrow x_1[d]$
$x_0^f \leftarrow M \cdot x_0 + (1 - M) \cdot x_1$
**return** $x_0^f$

---

## Appendix D. Vector Quantization

The study on the deep discrete representation is initiated in (Van Den Oord et al., 2017), the authors proposed VQ-VAE, an autoencoder with discrete latent representation. The

authors proposed a trainable codebook $e \in \mathbf{R}^{K \times D}$, where $K$ is the number of latent codes and $D$ is the dimension of each code vector. For an input $x$, encoder output $z_e(x) \in \mathbf{R}^{L \times D}$, let $e_x$ be the quantized vector of $x$, namely, for $i = 1, \ldots, L$,

$$e_{x,i} = \arg\min_k ||z_e(x)_i - e_k||_2, \tag{2}$$

In order to train $z_e(x)$, let $z_q(x)$ be the straight-through estimator (Bengio et al., 2013) of $e_x$ with respect to $z_e(x)$, i.e., $z_q(x) = z_e(x) + \text{sg}[e_x - z_e(x)]$, where sg stands for the stop gradient operator. Then $z_q(x)$ is passed through the decoder for reconstructions and the training objective is

$$L = \log p(x|z_q(x)) + ||\text{sg}[z_e(x)] - e_x||_2^2 + ||z_e(x) - \text{sg}[e_x]||_2^2. \tag{3}$$

Vanilla Variational Autoencoders (VAEs) are known to suffer from the problem of posterior collapse (Wang et al., 2021), wherein the latent space becomes excessively noisy, causing the decoder or autoregressive model to neglect the fine-grained details encoded in the latent representations. On the other hand, the authors empirically demonstrate that the learned discrete latent codes retain meaningful information, and varying combinations of these tokens produce distinctly different outputs. To enhance the quality of image generation, (Esser et al., 2021) introduced VQGAN, which integrates the training objective defined in Equation 3 with additional discriminator and perceptual loss terms.

However, from Equation 3 we see that code vectors nearest to $z_e(x)$ are further adjusted to be even closer to it, which consequently increases the likelihood that these vectors will be selected in subsequent training iterations. This dynamic leads to a low utilization rate of the codebook $e$, thereby limiting the representational capacity of the model. To address this limitation, (Zhu et al., 2024) introduced VQGAN-LC. Rather than initializing the codebook randomly, (Zhu et al., 2024) constructed a fixed codebook by extracting patch-level features from datasets characterized by rich texture information (e.g., ImageNet or FFHQ) using a robust pretrained backbone network such as CLIP. Employing the same training objective as described in Equation 3, the authors then trained a projector to align the codebook with $z_e(x)$. (Zhu et al., 2025) introduced SimVQ, which reparameterizes the latent codebook through a learnable linear layer. Their findings demonstrate that this straightforward modification effectively mitigates the issue of codebook collapse.

## Appendix E. Condition Flow Matching

Diffusion models (Ho et al., 2020; Dhariwal and Nichol, 2021) have attained state-of-the-art performance in image generation tasks. Nonetheless, these models typically assume that the source distributions follow a Gaussian form. Flow matching (Lipman et al., 2023; Liu et al., 2023; Albergo and Vanden-Eijnden, 2023; Tong et al., 2023) be interpreted as a generalization of diffusion models to accommodate arbitrary source distributions (Gao et al., 2024). (Tong et al., 2023) introduced conditional flow matching and reformulated (Lipman et al., 2023; Liu et al., 2023; Albergo and Vanden-Eijnden, 2023) within this framework. This subsection provides a concise overview of the conditional flow matching formulation.

Given a probabilistic density $p_0(x) \in \mathbb{R}^d$, we would like to construct a continuous family of probabilistic distributions $p_t(x), t \in [0, 1]$ from the solution of the following differential equation

$$dx = u(t,x)dt, \ u(t,x) \in C^1([0,1] \times \mathbb{R}^d, \mathbb{R}^d). \tag{4}$$

That is, let $\phi_t(x)$ be the solution of equation 4, and , we will construct $p_t$ as the push-forward measure $p_t := [\phi_t]_\#(p_0)$. The probability path $p : [0,1] \times \mathbb{R}^d \to \mathbb{R}$ such that $p(t,x) = p_t(x)$ is characterized as the famous continuity equation

$$\frac{dp}{dt} = -\nabla \cdot (p_t u_t), \tag{5}$$

where $u_t(x) := u(t,x)$, with initial condition $p_0$.

Given a sampler on the joint distribution of source and target distributions $q(z)$, a conditional probabilistic distribution $p_0(x|z)$, and a computable $u_t(x|z)$, the conditional flow matching objective is defined as

$$\mathcal{L}_{CFM}(\theta) = \mathbb{E}_{t,q(z),p_t(x|z)}||v_\theta(t,x) - u_t(x|z)||^2, \tag{6}$$

where $p_t(x|z) := [\phi_t]_\#(p_0)$, $\phi_t$ is the solution of differential equation $dx = u(t,x|z)dt$, and $v_\theta$ is the neural network to be trained. The reformulations of (Lipman et al., 2023; Liu et al., 2023; Albergo and Vanden-Eijnden, 2023) are presented in (Tong et al., 2023) Table 1. For example, for (Liu et al., 2023), $q(z)$ is the independent coupling $q(x_0)q(x_1)$, $u(t,x|z) = x_1 - x_0$, and $p_t(x|z) = t \cdot x_1 + (1-t) \cdot x_0$.

Let $\Pi$ be the set of probabilistic distributions whose marginals are $q_0 := q(x_0)$ and $q_1 := q(x_1)$, and let 2-Wasserstein distance between distributions $q(x_0)$ and $q(x_1)$ with respect to Euclidean metric defined as the following:

$$W_2(q_0, q_1) := \inf_{\pi' \in \Pi} \int_{\mathbb{R}^d \times \mathbb{R}^d} ||x - y||^2 d\pi'(x,y), \tag{7}$$

and let $\pi(x_0, x_1) \in \Pi$ be the distribution achieving the optimal in Equation 7. (Tong et al., 2023) proposed to use $q(z) := \pi(x_0, x_1)$, $u(t,x|z) = x_1 - x_0$, and $p_t(x|z) = \mathcal{N}(x|t \cdot x_1 + (1-t) \cdot x_0, \sigma^2)$. (Tong et al., 2023) call this method optimal transport CFM (OT-CFM) and faster training and inference compared with other baselines are demonstrated.

## Appendix F. Feature Interpolation

In this study, we explore the application of conditional flow matching to transform pathological images into normal counterparts and to quantify the differences between these image pairs. Conditional flow matching assumes that the final trajectory comprises points obtained through linear interpolation between data points. To validate the appropriateness of employing this method within the latent space of a vector quantized autoencoder, we perform interpolations between two distinct images in the pixel space and in the latent spaces of VAE, VQGAN, and OptVQ, respectively. We select one pathological image $x_0$ and one normal image $x_1$, and visualize their interpolations defined by $x_t = (1-t)x_0 + tx_1$ on $t = 0.2, 0.4, 0.6, 0.8$. The resulting interpolated images are presented in Fig. 9.

In the direct interpolation of two images, as illustrated in the first row of Fig. 9, numerous artifacts are evident, particularly at $t = 0.4$, the ventricles from the two images appear conflated, with noticeable artifacts surrounding the skull region, and the patches

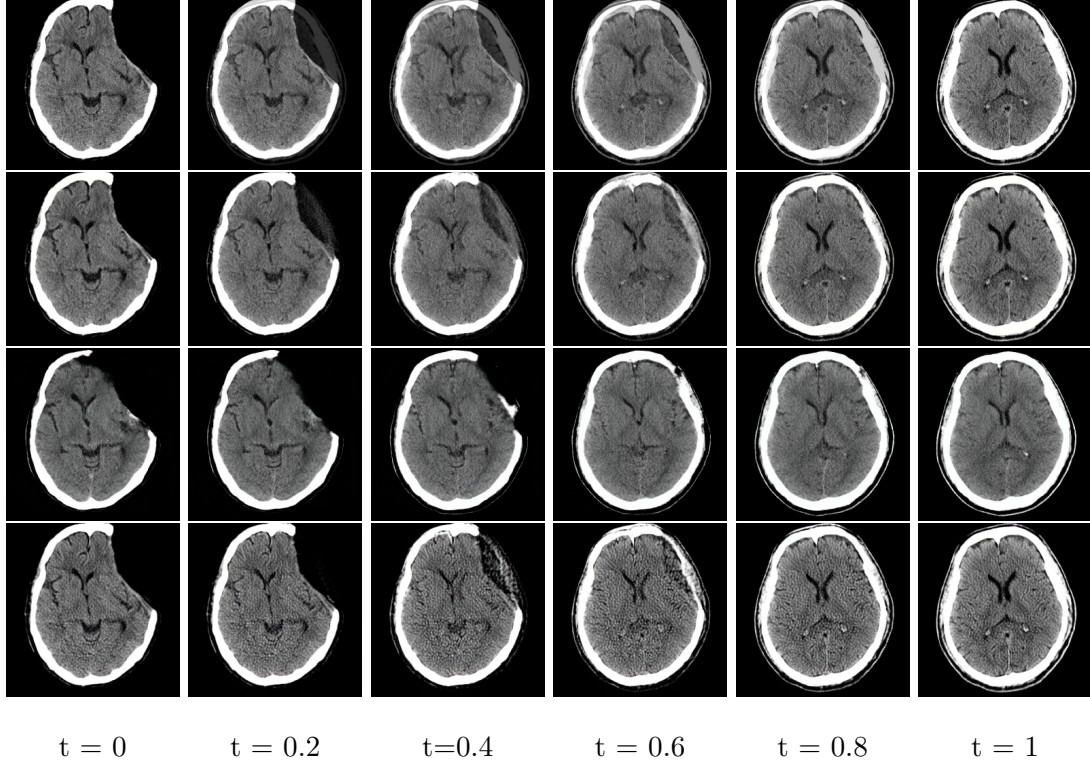

| | | | | | |
|---|---|---|---|---|---|
| t = 0 | t = 0.2 | t=0.4 | t = 0.6 | t = 0.8 | t = 1 |

Figure 9: Illustrations of feature interpolations of different methods. From above to below: image interpolation, VAE, VQGAN, OptVQ.

near the left frontotemporal area do not resemble those originating from a single coherent image. These issues are mitigated when interpolation is performed within the latent space of VAE. At $t = 0.4$, two right ventricles are present in the same image, and the VAE attempts to interpret features from both the left and right ventricles of the source image as a single left ventricle in the interpolated output. Additionally, no artifacts are observed around the skull in this case. VQGAN demonstrates remarkable interpolation performance, even in the absence of conditional flow matching. At every interpolation step, the generated ventricles appear realistic, with minimal artifacts. However, possibly due to limitations related to codebook utilization, the reconstruction capability is constrained, resulting in the omission of some fine-grained textures. Lastly, the OptVQ method seeks to balance these factors; although non-realistic reconstructions of ventricles persist at $t = 0.4$, the merging of the two ventricles from the source image into a single (albeit unrealistic) right ventricle is somewhat improved compared to the VAE-generated images. Furthermore, the reconstruction quality achieved by OptVQ surpasses that of VQGAN and is comparable to that of the VAE.

## Appendix G. Performances on the Variational Autoencoder

VAEs are frequently employed as the autoencoder architecture in subsequent diffusion or flow matching generative models (Rombach et al., 2022). In the present study, we train a VAE with a latent representation of dimensions $16 \times 16 \times 16$, followed by training a flow matching model on this latent space. As demonstrated in Fig. 13, the reversed disease progression can be generated in a smooth manner. Nevertheless, we observe that when using VAEs, CutMix-like augmentations can introduce artifacts into the generated images.

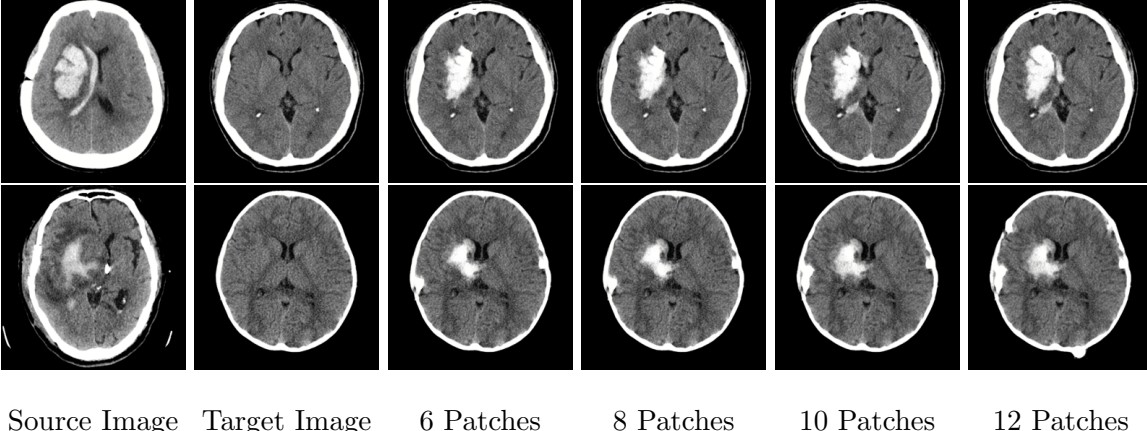

Source Image    Target Image    6 Patches    8 Patches    10 Patches    12 Patches

Figure 10: Illustrations of the effects of intensity shuffling on VAE with different numbers of patches.

Fig. 10 illustrates the impact of intensity shuffling. It is observed that the pixel values corresponding to the hemorrhage regions in the generated image are excessively elevated, while the surrounding edema is entirely omitted and fails to be reconstructed. When intensity shuffling is employed during the training of the flow matching model, as demonstrated in Fig. 14, a comparison with Fig. 13 reveals the emergence of numerous artifacts surrounding the brain skulls.

In the context of patch shuffling, as demonstrated in Fig. 11, while the reconstruction of the frontal horns is nearly flawless, the patches within the cropped areas exhibit inconsistencies when compared to those adjacent to the cropped regions. Notably, shadows and patches of high intensity are also produced.

## Appendix H. Performances with different vector quantizations

We also conducted the experiment described in Section 3.4 using SimVQ (Zhu et al., 2025), a method that enhances the performance and stability of VQGAN by incorporating a linear learnable layer into the latent codebook. The results are illustrated in Fig. 15. When compared to Fig. 12, we find the reconstructed images produced by SimVQ and VAE exhibit a high degree of similarity. Notably, while both SimVQ and VAE tend to reconstruct darker patches within the masked regions, OptVQ generates patches with pixel values that approximate the average intensities of the original images.

## Appendix I. More Results on the Intensity Shuffling and Patch Shuffling

Fig. 16 presents the generated images obtained with and without the application of patch shuffling during inference. Our findings indicate that the proposed models accurately restore the frontal horns to the optimal midline position, including cases exhibiting minimal midline displacement, as demonstrated in the third and fourth rows of Fig. 16. Furthermore, when

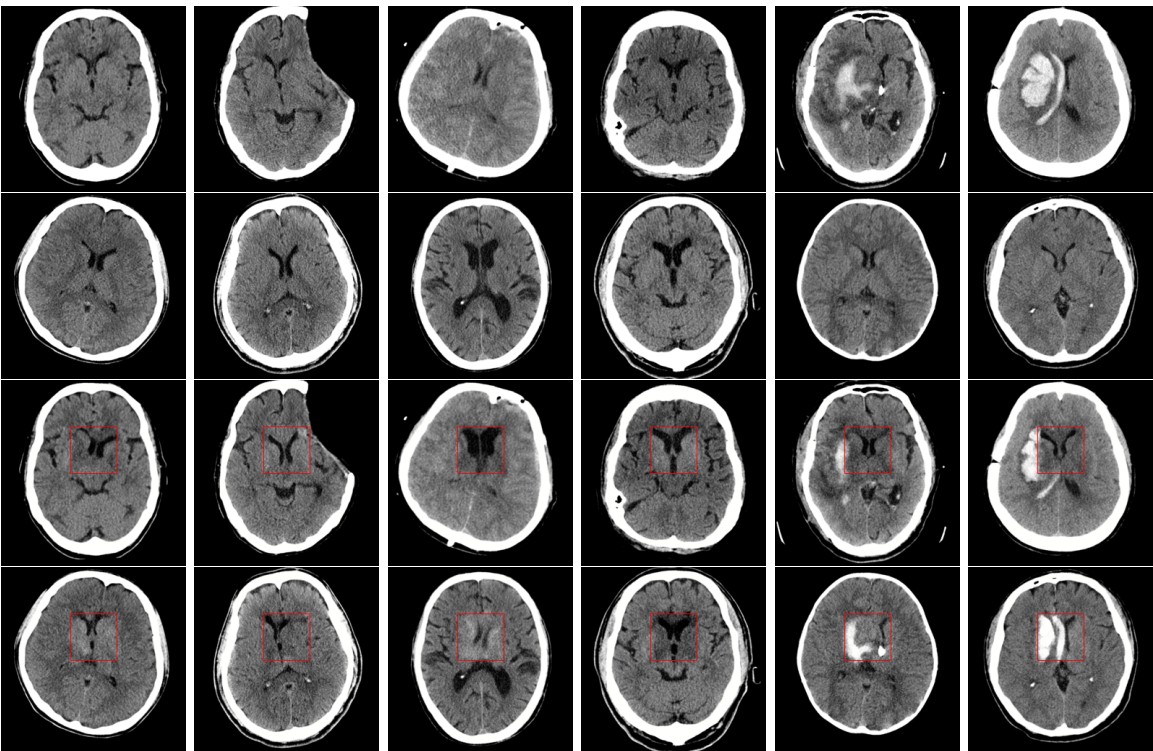

Figure 11: Illustrations of the patch shuffling on VAE. From up to below: source images, target images, source images after the patch shuffling with target images, target images after the patch shuffling with source images.

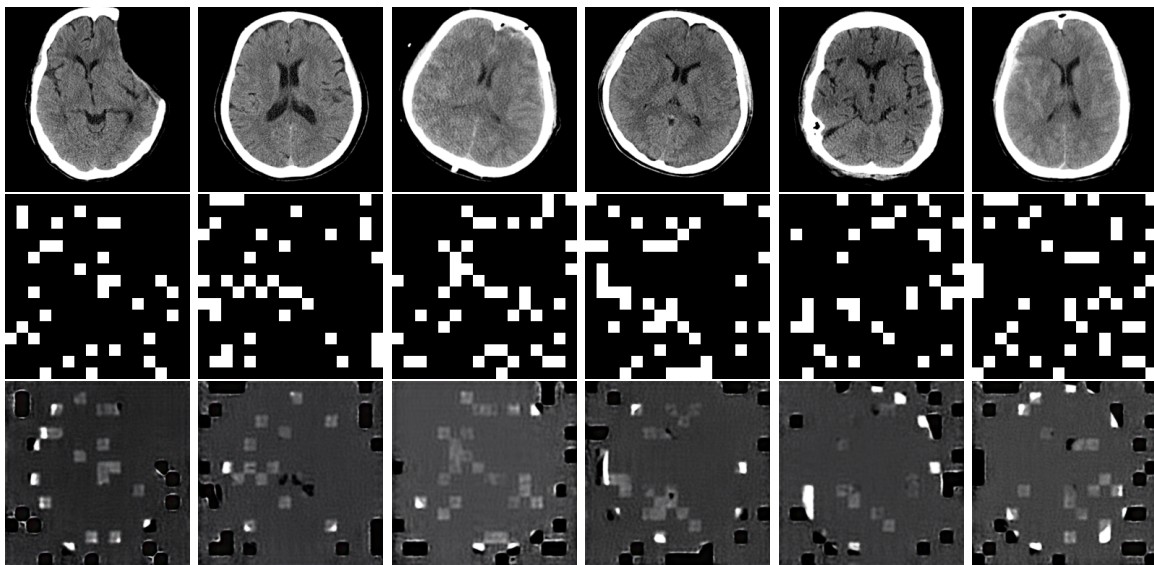

Figure 12: Illustrations of the reconstructions of masked images. From up to below: original images, masks, reconstructed images by VAE.

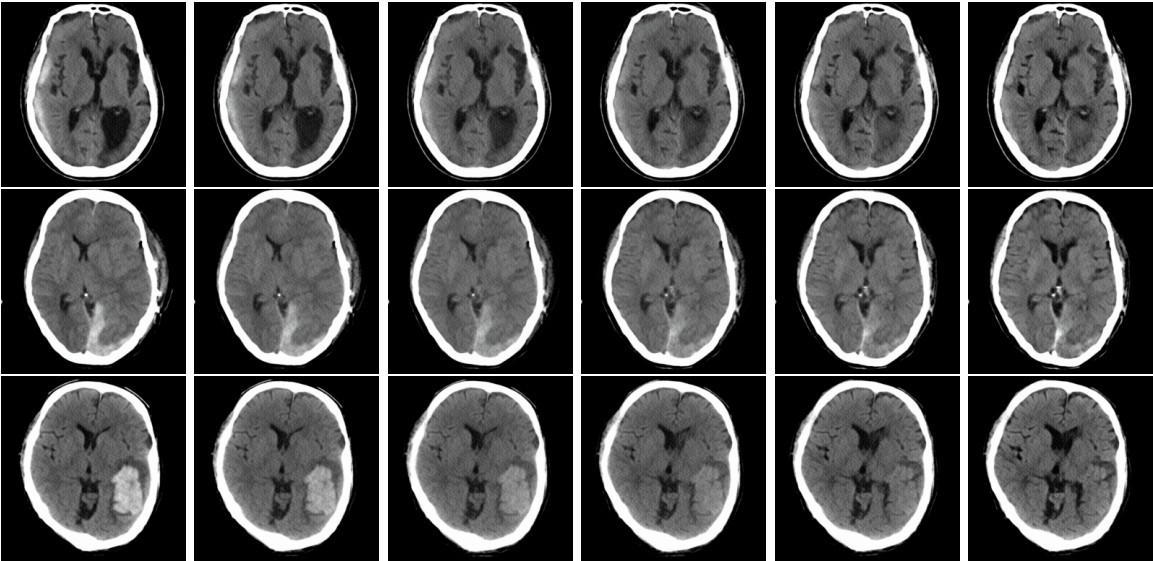

Figure 13: Illustrations of the generations of reverse disease progress on the latent space of VAE.

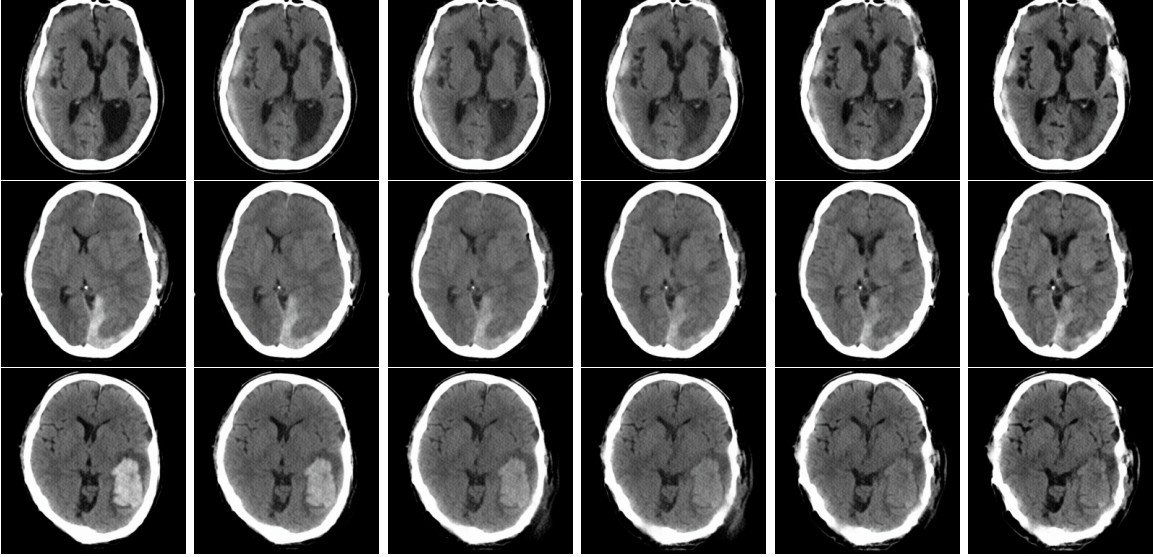

Figure 14: Illustrations of the generations of reverse disease progression trained with intensity shuffling augemented data on the latent space of VAE.

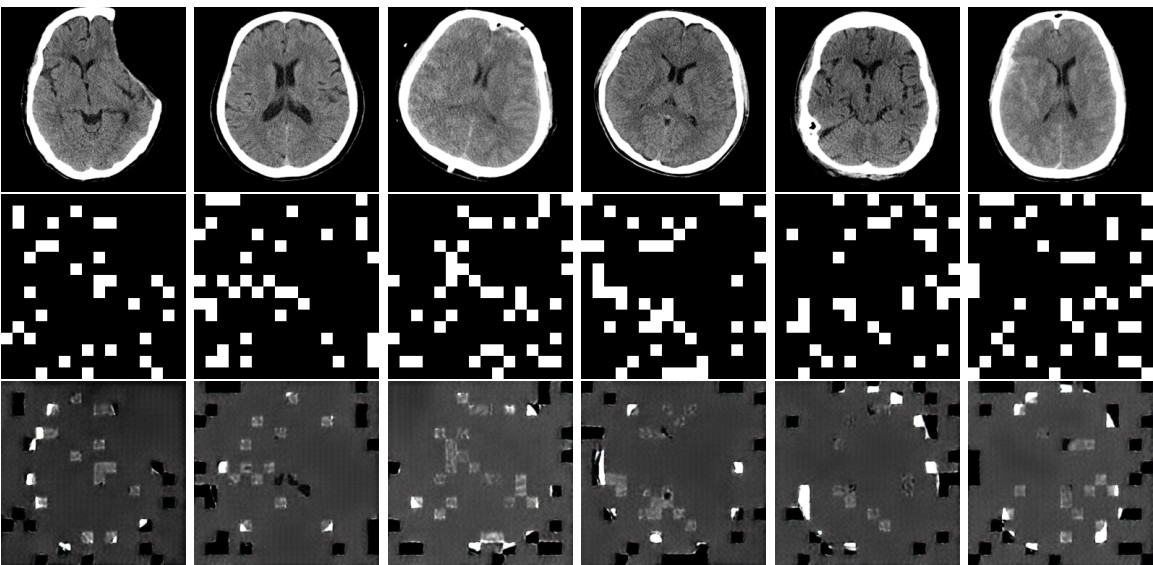

Figure 15: Illustrations of the reconstructions of masked images. From up to below: original images, masks, reconstructed images with SimVQ.

patch shuffling is applied during inference, all unrelated regions remain unaltered throughout the reversal of disease progression.

## Appendix J. Failure Case Study

Fig. 17 depicts the most illustrative examples of unsuccessful cases. In the first two columns, although the flow matching model successfully reconstructs plausible normal images, the performance of the diffeomorphic registration is significantly compromised by the presence of edema in the right hemisphere, whose pixel intensity closely resembles that of the frontal horns. Consequently, the edema is incorrectly deformed to form the right frontal horn. In the third and fourth columns, the frontal horns in the source images are notably small, leading to their deformation toward adjacent pixels rather than the intended final ventricular structures.

In the fifth column, the left frontal horn of the source image is positioned near the septum pellucidum in the generated image, resulting in its deformation toward the septum pellucidum rather than toward the left frontal horn of the generated image. In the final column, it is likely that the substantial spatial separation between the frontal horns of the source and generated images prevents the frontal horns of the source image from being accurately warped to the corresponding regions in the generated image.

## Appendix K. Hyperparameters

For hyperparameters, most of time we followed the official implementations of (Esser et al., 2021), (Rombach et al., 2022) , and (Zhang et al., 2024), respectively. For VQGAN model in (Chih-Chieh and Chang-Fu, 2025), we further increased the codebook size from 1024 to

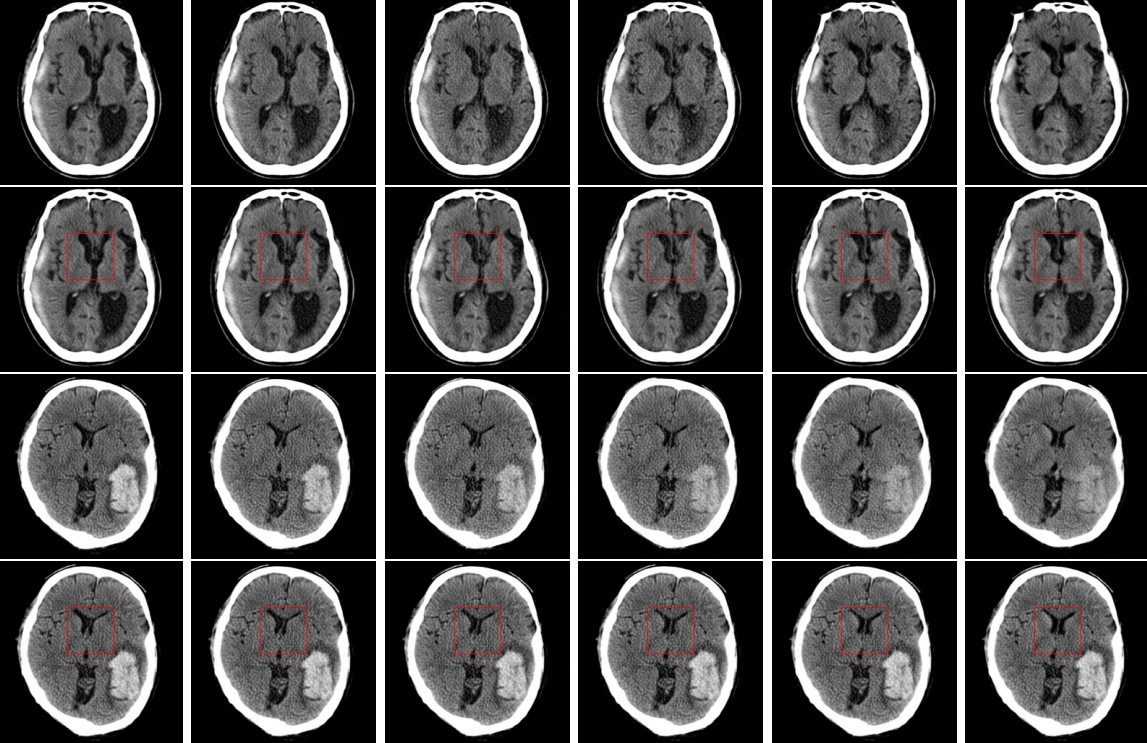

Figure 16: Illustrations of the generations of reverse disease progression. The first row and third row illustrate the flow matching model trained with the intensity shuffling, and the second and the fourth row represent the input images augmented by the generated images by using the patch shuffling during the inference time.

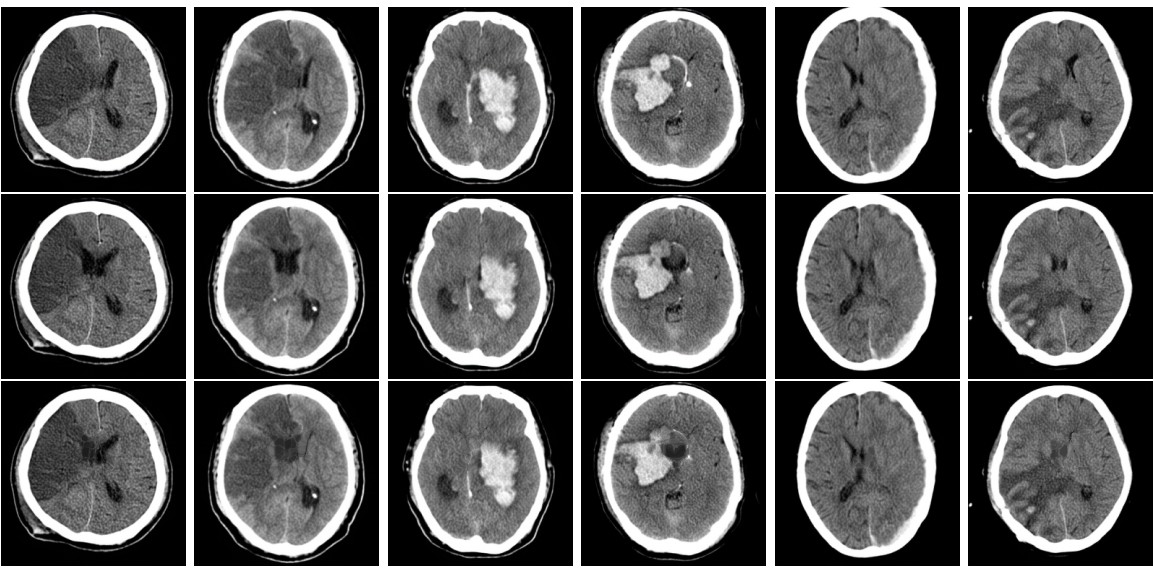

Figure 17: Illustrations of failure cases. From above to below: original images, reconstructed images, warped images.

4096, and replaced the self attention layers with oriented 1D kernels (Kirchmeyer and Deng, 2023). For training hyperparameters of OT-CFM, we followed the official implementation of (Tong et al., 2023): We use Adam optimizer with learning rate $2e-4$, and train for 20000 steps. Detailed hyperparameters are provided in Table 3. Regarding patch shuffling, since we work with rigidly registered images, the cropped region was set to [64: 128, 96:160].

Table 3: Hyperparameters in our experiements.

| Architecture | VQGAN | KL | OptVQ |
|---|---|---|---|
| Codebook Size | 4096 | None | 16384 |
| Latent Size | $16 \times 16$ | $16 \times 16$ | $16 \times 16 \times 4$ |
| Latent Embedding Size | 256 | 16 | 64 |
| Attention Type | Orient 1D kernel | Self Attention | Self Attention |
| Attention Layers | [64, 32 , 16] | [16] | [16] |
| Latent Flow Matching Model | UNet/UNetWarp | UNet | UNet/UNetWarp |
| Downsampling Layers | 3 | 3 | 3 |
| Initial Channel Size | 256 | 128 | 128 |
| Channel Multiplicty | [1, 2, 2] | [1, 2, 2, 4] | [1, 2, 2, 4] |
| Attention Layers | None | None | None |
| Training Steps | 80K | 20K | 20K |

