# OpenReview forum: "Vector Quantization for Reversed Disease Progression: Further Investigations"
_MIDL.io/2026/Conference — MIDL 2026 Poster_

### Official Review · Reviewer_AGfq · 2026-01-13

**Confidence:** 3
**Preliminary Rating:** 3
**Final Rating:** 4

**Summary:**

This paper proposes a method for quantifying midline shift in brain CT scans by simulating "reversed disease progression" (transforming pathological images into their healthy counterparts). Building on previous work that utilized conditional flow matching, this study investigates the use of OptVQ (Sinkhorn-based quantization) to improve the latent representation of the autoencoder. To address specific challenges in clinical data, the paper introduces two data augmentation strategies: intensity shuffling and patch shuffling.

**Strengths:**

1. The application of Sinkhorn-based quantization (OptVQ) to the specific problem of medical image generation is interesting. The paper provides some evidence (e.g., masked image reconstruction) that this approach preserves spatial correspondence better than standard VQGAN, which is a valuable insight.
2. The overall goal of reversed disease progression aligns well with the need for interpretable AI in clinical settings.

**Weaknesses:**

1. The introduction discusses clinical challenges broadly (unlabeled data, annotation effort) but fails to clearly specify the primary task, midline shift quantification, early on. The reader is left to infer the specific problem being addressed, which only becomes clear when discussing related work or experimental setups.
2. The paper suffers from pacing issues. Section 2 (Related Work) spans 2.5 pages, which is excessive for a conference paper and delays the methodological contribution. The conclusion is also long and lacks the concise expected of a concluding section.
3. The paper lacks a comprehensive overview figure illustrating the proposed framework (OptVQ + Flow Matching + Shuffling).
4. Table 1 lacks critical baseline comparisons. While it reports the Mean Absolute Error (MAE) for the proposed method ("w/o" and "with" patch shuffling), but it does not include results from the authors' own previous work (Chih-Chieh and Chang-Fu, 2025), which is the basis for this further investigation, and the state-of-the-art methods mentioned in the related work.

**Detailed Comments:**

1. The architectural specifics of the core models are missing. While "conditional flow matching" and "OptVQ" are mentioned, the authors do not specify the backbone of the flow matching network (e.g., U-Net vs. DiT, depth, width) or the specific configuration of the OptVQ autoencoder (e.g., codebook size K, embedding dimension D).

**Justification Of Final Rating:**

The authors have successfully addressed my concerns regarding the framework figure, comparison tables, and overall writing quality in the comment and revised paper.  I have raised my score to a Weak Accept

**Justification Of The Preliminary Rating:**

This paper introduces a technically sound extension to reversed disease progression using OptVQ and novel shuffling strategies. However, validation is insufficient for a full conference paper due to a critical lack of baseline comparisons. Furthermore, missing architectural definitions and framework figures hinder reproducibility. Despite the methodological promise, I recommend a borderline and will reconsider in rebuttal.

**Questions To Address In The Rebuttal:**

Please provide quantitative baseline comparisons (MAE) against previous work and state-of-the-art methods, along with specific architectural details and a framework diagram to ensure reproducibility.

---

> ### Author Response · Authors · 2026-01-21
> **Quantitative results are provided in Table 1 and Table 2**
>
> Dear Reviewer AGfq:
>
> Thank you for your valuable suggestions. In response to your concerns, we provide the following clarifications:
>
> * The architectural specifics of the core models are missing.
>
> The hyperparameters are now summarized in Appendix K. Briefly, we use UNet without attention layers in the ResBlocks. In our previous work, we replaced the attention layers of the VQGAN model with oriented 1D kernels due to concerns about potential loss of spatial information. However, as demonstrated in Section 3.4 of this work, spatial consistency is preserved even with attention layers. Therefore, we adopt the default settings of VAE and OptVQ in this study.
>
> * The reader is left to infer the specific problem being addressed, which only becomes clear when discussing related work or experimental setups.
>
> We have modified some sentences in the first paragraph of p. 2; we hope it can make the article more focused on the topic.
>
> * The paper suffers from pacing issues.
>
> Now we have removed parts of the contents from Appendix A and Appendix D, reducing Section 2 to 1.5 pages.
>
> * Please provide quantitative baseline comparisons (MAE) against previous work and state-of-the-art methods, along with specific architectural details and a framework diagram to ensure reproducibility.
>
> The findings are presented in Table 1 and Fig. 7, and the checkpoints along with the network architectures can be accessed on our GitHub page. Our procedure is outlined in Fig. 6 of our updated manuscript. Additionally, we have included a straightforward experiment to demonstrate that our methods are relatively unbiased, with the results detailed in Table 2.
>
> The comprehensive details are located in the second paragraph on page 10 (highlighted in yellow). We remind readers of the findings from our earlier research. Additionally, to produce more detailed images, which is a primary goal of our study, we experimented with the network architecture from our previous work under OptVQ, which we call UNetWarp. Surprisingly, for quantitative data, OptVQ+UNetWarp delivered the highest performance. Regarding intensity shuffling, as shown in the first three rows of Fig. 7, even normal images exhibit slight alterations around the frontal horns in the model trained with intensity shuffling. Meanwhile, OptVQ+UNet and OptVQ+UNetWarp nearly flawlessly reconstructed the normal images. We believe this is why the MAE for normal images is higher than in our previous work.
>
> Nonetheless, we wish to highlight that a key aim of this study is to produce unbiased reversed disease progressions. As shown in the final three rows of Fig. 7, OptVQ+UNet and VAE+UNet fail to maintain the hematographic regions. OptVQ+UNetWarp retains some of these areas, while the model with intensity shuffling (along with patch shuffling) preserves the regions flawlessly.
>
> We also conduct a simple experiment to estimate the mIoUs of  hemographic regions in both the source and generated images, illustrating that our proposed approach (intensity shuffling combined with patch shuffling) retains more hemographic regions compared to other techniques. The findings are detailed in Table 2.

---

### Official Review · Reviewer_Qn1V · 2026-01-16

**Confidence:** 4
**Preliminary Rating:** 3
**Final Rating:** 4

**Summary:**

This paper studies simulation of reversed disease progression using a vector-quantized autoencoder with Sinkhorn-based quantization (OptVQ). The authors use head MRI data and midline shift quantification as the primary use case, aiming to transform pathological images into “normal” images by simulating reverse disease progression. The paper further proposes two latent-space augmentations, intensity shuffling and patch shuffling to improve image generation process
Overall, the paper is interesting, particularly the use of OptVQ for reverse disease progression (midline shift distance estimation) and the idea of patch/intensity shuffling in latent space. However, the current experimental evidence does not convincingly demonstrate performance compared to baseline models, and several key quantitative analyses are missing.

**Strengths:**

* The paper addresses an important and clinically relevant problem, simulating reversed disease progression for neuroimaging pathologies.
* Performing Sinkhorn-based quantization in latent space is technically interesting and potentially efficient.
* The work extends prior work on conditional flow matching by integrating it with a Sinkhorn-quantized latent representation.
* The proposed latent-space shuffling augmentations (intensity and patch shuffling) are conceptually interestiong and could be broadly applicable.
* The experiments suggest that patch shuffling improves midline shift prediction error (MAE), indicating the augmentation may affect representation learning.

**Weaknesses:**

* The MAE reported in Table 1 appears high and worse than prior results using VQGAN with a linear classification head. This needs clarification.
* A substantial portion of the paper explains OptVQ / Sinkhorn-based quantization, which has been proposed before, reducing the perceived novelty.
* The paper lacks thorough ablation studies isolating the effects of (i) Sinkhorn-based quantization vs. standard nearest-prototype assignment, (ii) intensity shuffling, and (iii) patch shuffling.
* Section 3.1 is titled “Fair” Reversed Disease Progression. The meaning of “fair” is unclear. This should be defined precisely or renamed.
* Intensity shuffling is highlighted as a key technique, but there is no direct quantitative evaluation showing when/how it helps, or whether it improves robustness/generalization.

**Detailed Comments:**

* The title emphasizes Sinkhorn-based quantization, but the experiments do not sufficiently isolate the benefit of Sinkhorn-based quantization compared with a simpler quantization baseline (e.g., nearest codebook assignment). A focused comparison is needed to justify the framing.
* The abstract/introduction emphasize interpretability for clinicians, but the paper does not explain how the proposed method improves interpretability, nor does it provide interpretability-focused experiments or analyses. If interpretability is not a core contribution, the motivation should be reframed.
* The paper claims to address data imbalance, but it is not clearly shown how intensity/patch shuffling specifically mitigates imbalance.

**Justification Of Final Rating:**

The authors have addressed all issues raised, added relevant baseline comparisons, and conducted additional experiments that demonstrate the benefits of their method. I have therefore revised my recommendation to weak accept.

**Justification Of The Preliminary Rating:**

This paper is overall well written. The authors propose an interesting method; however, the paper does not quantitatively demonstrate that the MAE for midline shift distance is lower, nor does it compare the proposed model (with Sinkhorn-based quantization and the augmentation technique) against baseline methods.

**Questions To Address In The Rebuttal:**

* Why are the Table 1 results worse than prior work ([1], VQGAN + linear head)?
* If intensity shuffling is applied aggressively, does it risk weakening the model’s ability to preserve clinically important hyperintense/hemorrhagic regions in the reverse-progression simulation?
* How is the position of patches defined for the patch augmentation?
* Please provide additional evidence about the efficacy of patch shuffling.
* The authors should briefly describe how conditional flow matching is applied in their model. Otherwise, it is unclear how it is used.
[1] Chen, Chih-Chieh; Kuo, Chang-Fu. Weakly-supervised midline shift quantification through simulating the reversed disease progression. MIDL 2025 (Short Paper), 2025.

---

> ### Author Response · Authors · 2026-01-21
> **More results are summarized in Table 1, Table 2, and Fig. 7**
>
> We sincerely thank for your informative suggestions. We summarize our responds as follows:
>
>  * Why are the Table 1 results worse than prior work ([1], VQGAN + linear head)?
>
> We have added several results to Table 1, including flow matching models without any augmentations. Interestingly, as shown in the first three rows of Figure 7, applying intensity shuffling consistently causes slight changes around the frontal horns in normal images, which may explain the higher MAE observed for these images. During each training iteration, we shuffle patches with the top 10-20 highest intensity values; however, we did not set a threshold for intensity values, meaning that patches with normal intensity might also be shuffled. We believe this could be the cause, although we have not fully investigated it yet. On the other hand, as seen in the last three rows of Figure 7, without any regularization, pathological features such as hematomas disappear in the generated images. This phenomenon is further quantified in Table 2. For more details, please refer to the third paragraph (highlighted in yellow) of our updated manuscript.
>
> *  If intensity shuffling is applied aggressively, does it risk weakening the model’s ability to preserve clinically important hyperintense/hemorrhagic regions in the reverse-progression simulation?
>
> Although the quantitative results are lower, intensity shuffling actually helps preserve hemorrhagic regions. In our experience, for flow matching models (operating in the latent space of VAE and OptVQ) without any regularization, most hemorrhagic regions disappear after 10K iterations. In contrast, when using intensity shuffling, most of these regions remain intact even after 30K iterations. As shown in Fig. 16, some features within these regions are lost and some unrelated features are altered. This is why we introduced patch shuffling—to keep features unchanged in areas that are not of interest.
>
> * How is the location of patches determined for patch augmentation?
>
> We are using CT images that have been rigidly registered to a single fixed template. This allows us to estimate the position of the frontal horns, which are used to measure the midline shift distance. For images sized 256x256, we select the region [64:128, 96:160].
>
> * The authors should provide a brief explanation of how conditional flow matching is utilized in their model.
>
> We have outlined the process in Figure 6 of our revised manuscript.
>
> * The experiments do not sufficiently isolate the benefit of Sinkhorn-based quantization compared with a simpler quantization baseline
>
> We agree with the reviewer. In the Appendix, we provide empirical evidence showing that Cutmix-like operations are not appropriate for VAEs. However, the comparison between VQ models is insufficient. Therefore, we have revised the motivation in the last paragraph of page 2 (highlighted in yellow).
>
> * The paper lacks thorough ablation studies isolating the effects of (i) Sinkhorn-based quantization vs. standard nearest-prototype assignment, (ii) intensity shuffling, and (iii) patch shuffling.
>
>  Regarding MAE, the results are summarized in Table 1. To measure the degree of bias in the outputs. Please see Section 4 for more details.
>
> * The meaning of “fair” is unclear.
>
> We now use the word "unbiased."
>
> * The paper does not explain how the proposed method improves interpretability.
>
> We adopt the concept of interpretability from [1], that is, the reasoning process is understandable by Human. Consequently, we suggest generating the reversed disease progression and measuring the differences between the source and target images. This information has now been included in the second paragraph of Section 1.
>
> * It is not clearly demonstrated how intensity or patch shuffling specifically addresses the imbalance.
>
> In our study, the imbalance arises from the hyperintense or hemorrhagic regions; therefore, we augment patches located in these high-intensity areas to match the target during the training of the flow matching model. Additionally, we performed a quantitative analysis, which is presented in Table 2 of the revised manuscript.
>
> * Intensity shuffling is highlighted as a key technique, there is no direct quantitative evaluation showing when/how it helps.
>
> We have now conducted a quantitative analysis, which is shown in Table 2 of the updated manuscript.
>
> *  the paper explains OptVQ / Sinkhorn-based quantization, which has been proposed before, reducing the perceived novelty.
>
> We agree with the reviewer. The main contribution of our work is to propose several methods to reduce the bias, and investigate the trade-off on visual and quantitative information. As illustrated in Fig. 11 and Fig. 14, our method does not work well on VAE model, this gives us reasons to further investigate VQ-based architectures.
>
> [1] Chen et al. This looks like that: deep learning for interpretable image recognition.

---

> > ### Comment · Reviewer_Qn1V · 2026-01-29
> > **Discussion**
> >
> > -	The authors have addressed all issues raised by the reviewers.
> > -	They have compared their method against relevant baseline approaches and added additional experiments, which improves the overall quality of the paper.
> >
> > There are a few issues that should be addressed:
> > -	The quantitative improvements for MAE are marginal; however, the paper’s main contribution appears to be the preservation of pathological features rather than maximizing standard performance metrics.
> > -	The idea of including an overview figure (Figure 6) is valuable; however, the current version is difficult to interpret and should be revised for clarity. For example, it is unclear what inputs are fed into the U-Net and what the Warp block outputs (and how these outputs are subsequently used). A more descriptive caption and clearer labeling of inputs/outputs (including tensor shapes or key intermediate representations) would likely resolve these ambiguities.
> > -	The idea behind the experiments in Table 2 is godd, but the manuscript should clearly describe how the samples were selected. The statement “We selected CT images containing large acute hematomas…” feels somewhat arbitrary without further justification and could introduce selection bias. In addition, the IoU reported for I + P (0.99) seems very high and warrants clarification.

---

> ### Author Response · Authors · 2026-01-30
> **Further clarification about Table 2**
>
> Dear Reviewer Qn1V,
>
> Thanks for your constructive information. We would like to provide additional explanation here as we are currently unable to make changes to the manuscript.
>
> * The quantitative improvements for MAE are marginal; however, the paper’s main contribution appears to be the preservation of pathological features rather than maximizing standard performance metrics.
>
> We concur with the reviewer.
>
> * it is unclear what inputs are fed into the U-Net and what the Warp block outputs (and how these outputs are subsequently used). A more descriptive caption and clearer labeling of inputs/outputs (including tensor shapes or key intermediate representations) would likely resolve these ambiguities.
>
> We apologize for the confusion.  We would like to incorporate more details regarding UNetWarp in the following flow chart:
>
> Input (256 x 16 x 16) &rarr; UNet &rarr; (  S_Out (256  x 16 x 16) , D_Out (16 x 16 x 2)  ) &rarr; F.grid_sample(S_Out,  meshgrid (16 x 16 x 2) + D_Out ) (256 x 16 x 16),
>
> where the meshgrid is the 2-D grid coordinates of the latent spaces, normalized to (-1,1).
>
>
> * The statement “We selected CT images containing large acute hematomas…” feels somewhat arbitrary without further justification and could introduce selection bias. In addition, the IoU reported for I + P (0.99) seems very high and warrants clarification.
>
> We apologize for the incomplete information. For Table 2, we attempt to provide quantitative results to validate our observations in the fourth row of Fig. 7.   Since we do not have segmentation labels for acute hematomas, we select these slices by the following criterion: Our PNG images are sized 256 x 256,  with HU values range from 0 to 80.  The slice will be used in this experiment if there are more than 3000 pixels with values between 48 and 72 (to exclude the brain skulls). Then we create the masks with true values on these pixels.
>
> For the target images (images that pass through the flow matching model), we relax the constraint by creating masks with true values on pixels with values greater than 40. Outputs with slightly lower HU values are acceptable. That is perhaps why the IoU is so high. We also summarize the findings using the initial criterion (HU value 48) as follows:
>
> |  KL | OptVQ  |  (I)   | (I) + (P) |  (W)   | (W) + (P)  |
> |  ----  | ----  | ----  | ----  |----  | ----  |
> | 0.43 &pm;  0.02 | 0.48 &pm;  0.02 | 0.64 &pm;  0.01 | 0.95 &pm;  0.05 | 0.57 &pm;  0.19 | 0.92 &pm;  0.01 |

---

### Official Review · Reviewer_bztb · 2026-01-16

**Confidence:** 4
**Preliminary Rating:** 3
**Final Rating:** 4

**Summary:**

The paper introduces a method for reversed disease progression generation for medical images, by replacing the autoencoder with OptVQ vector-quantized autoencoder using Sinkhorn-based quantization. To improve the stability of the model under a limited data regime, the authors propose intensity shuffling and patch shuffling based on empirical results.

**Strengths:**

1. The formulation of the problem is well-motivated by the clinically meaningful needs for interpreting diseases.
2. The paper proposes simple but novel augmentations (i.e., intensity and patch shuffling), and verifies their effectiveness through quantitative and qualitative evaluation.
3. The paper includes comprehensive qualitative analysis.

**Weaknesses:**

1. The quantitative improvement is marginal with patch shuffling, adding the confidence interval is needed.
2. The number of tasks and datasets is limited. Therefore, the generalization of the method remains a concern.

**Detailed Comments:**

See Strengths and Weaknesses

**Justification Of Final Rating:**

I really appreciate the author's detailed feedback, as well as the additional experiments and clarifications provided during the rebuttal. The paper is more convincing after the clarification, and I am happy to raise my score.

**Justification Of The Preliminary Rating:**

The paper proposes novel data augmentation methods based on empirical studies. However, the quantitative improvement is marginal given no confidence interval and the limited number of the datasets and tasks leave a concern whether the method is generalize well.

**Questions To Address In The Rebuttal:**

Adding confidence intervals and potentially more datasets and tasks.

---

> ### Author Response · Authors · 2026-01-21
> **Confidence intervals are provided.**
>
> Dear Reviewer bztb,
>
> We sincerely appreciate your valuable suggestions. We address all your concerns as follows:
>
> - The quantitative improvement is marginal
>
> We have now included confidence intervals in Tables 1 and 2 of the revised manuscript. Additionally, we have incorporated several different baseline methods to better demonstrate the strengths and weaknesses of our approach. In summary, we acknowledge that the quantitative improvements are modest. In fact, in our reproduced experiments using the flow matching model without any augmentation, the MAE was even lower. However, we would like to emphasize that the primary objective of this work is to generate unbiased trajectories. As shown in Figure 7 of the revised manuscript, pathological features such as hematomas often disappear in the generated images. Our proposed methods—intensity shuffling and patch shuffling—effectively address this problem. For a detailed discussion, please refer to the third paragraph of Section 4 (highlighted in yellow). Additional illustrations of intensity shuffling and patch shuffling can be found in Figure 16.
>
> We have also conducted experiments demonstrating that our proposed methods yield more unbiased images. The quantitative results are summarized in Table 2.
>
> - The confidence interval is needed.
>
> The confidence intervals are now included in Table 1 and Table 2. We appreciate the reviewer's suggestion.
>
> - The number of tasks and datasets is limited. Therefore, the generalization of the method remains a concern.
>
> In this study, we primarily focus on the weakly supervised measurement of midline shift distances and the unbiased generation of reversed disease progressions. We acknowledge the reviewer's point that the number of tasks and datasets is limited. However, with the goal of broader generalization, we strive to present our observations as generally as possible. For instance, in Section 3.4, we explore the spatial relationship between input image patches and the corresponding latent vectors of the autoencoders at their specific locations, a topic we found lacking in existing literature. Additionally, in Appendix G, we provide empirical evidence showing that the proposed intensity shuffling and patch shuffling methods do not perform well in the latent space of the VAE. Although VAE is currently the predominant option for latent diffusion and flow matching models, our findings may serve as the motivations to further investigate the applications of VQ models.

---

### Author Rebuttal · Authors · 2026-01-21

**Rebuttal:**

We sincerely appreciate all the reviewers for their helpful feedback. We have revised significant portions of our manuscript, incorporating various baselines (including VAE and the network architecture from our previous study) and have summarized the results in Table 1, Table 2, and Figure 7. Regarding the mean absolute errors (MAEs), UNetWarp, the method from [1], achieves the best performance. However, our goal in this work is to generate an unbiased trajectory, and as shown in Figure 7, unrelated pathological features tend to disappear during the generation process. To address this, we propose several solutions in Section 3: intensity shuffling, patch shuffling, and the UNetWarp architecture from [1]. The first two approaches trade some precision to maintain feature integrity, while the third achieves the lowest MAE and preserves some features.

To better assess the level of bias in the generated images, we compute the mean Intersection over Union (mIoU) between the hemorrhagic region masks of the original images and those produced by different flow matching models. The results are presented in Table 2. As expected, our proposed methods maintain the hemorrhagic regions more effectively than flow matching models without any regularization.

Regarding our contributions, this work does not introduce new architectures; instead, we present several observations that may have been previously overlooked, summarized as follows:

1. We suggest applying the flow matching model (as opposed to the diffusion model) within the latent spaces of VQ models (rather than VAE). We believe this approach is not merely a different variant but that it is valuable to explore how various methods perform on similar tasks. Additionally, while most studies aim to minimize the number of iteration steps, we try to examine the outputs at each step to see the reversed disease progression.

2. We choose OptVQ as our VQ model. Compared to VAE, we observe that OptVQ is more robust when handling images without midline shift (as shown in the first three rows of Fig. 7).

3. To produce unbiased trajectories, we introduce CutMix-like augmentations applied to the latent features. We discovered that OptVQ is well-suited for this purpose, whereas using VAE can lead to artifacts (see Fig. 11 in Appendix G).

We sincerely thank all the reviewers once again.

[1] Chen, Chih-Chieh; Kuo, Chang-Fu. Weakly-supervised midline shift quantification through simulating the reversed disease progression.

**Supporting Material:**

/attachment/df9f12dde73b3e8602aed52c85331fc005969928.pdf

---

### Meta-Review · Area_Chair_McxW · 2026-02-07

**Recommendation:** Accept (Poster)
**Confidence:** 2

**Metareview:**

The paper introduces several strategies for improving disease progression image generation. Although it was initially framed as a general method, the proposed approach appears to be highly specialized for estimating midline shift and requires integration with a registration algorithm. Nevertheless, the paper received positive final ratings. The authors were actively engaged during the discussion period, and the primary concerns raised by the reviewers were adequately addressed through additional experiments and clarifications provided in the rebuttal.

---

### Decision · Program_Chairs · 2026-02-13

Accept (Poster)